# A new topology of the HK97-like fold revealed in *Bordetella* bacteriophage by cryoEM at 3.5 Å resolution

Xing Zhang[1], Huatao Guo[2], Lei Jin[2], Elizabeth Czornyj[2], Asher Hodes[2], Wong H Hui[1], Angela W Nieh[2], Jeff F Miller[1,2,3], Z Hong Zhou[1,2,3]*

[1]California NanoSystems Institute, University of California, Los Angeles, Los Angeles, United States; [2]Department of Microbiology, Immunology, and Molecular Genetics, University of California, Los Angeles, Los Angeles, United States; [3]Molecular Biology Institute, University of California, Los Angeles, Los Angeles, United States

**Abstract** Bacteriophage BPP-1 infects and kills *Bordetella* species that cause whooping cough. Its diversity-generating retroelement (DGR) provides a naturally occurring phage-display system, but engineering efforts are hampered without atomic structures. Here, we report a cryo electron microscopy structure of the BPP-1 head at 3.5 Å resolution. Our atomic model shows two of the three protein folds representing major viral lineages: jellyroll for its cement protein (CP) and HK97-like ('Johnson') for its major capsid protein (MCP). Strikingly, the fold topology of MCP is permuted non-circularly from the Johnson fold topology previously seen in viral and cellular proteins. We illustrate that the new topology is likely the only feasible alternative of the old topology. β-sheet augmentation and electrostatic interactions contribute to the formation of non-covalent chainmail in BPP-1, unlike covalent inter-protein linkages of the HK97 chainmail. Despite these complex interactions, the termini of both CP and MCP are ideally positioned for DGR-based phage-display engineering.

**\*For correspondence:**
Hong.Zhou@UCLA.edu

**Reviewing editor**: Stephen C Harrison, Harvard Medical School, United States

## Introduction

Capsid proteins of non-enveloped viruses fall, so far, into three major structural classes: the β-jellyroll, the HK97 fold, and the fold of dsRNA-virus shell proteins (*Bamford et al., 2005*; *Oksanen et al., 2012*). The RNA bacteriophage MS2 subunits have a fourth structure, not yet found in eukaryotic viruses (*Valegard et al., 1990*). The HK97 fold is present not only in a large number of dsDNA bacteriophage capsids, including the T-phages, lambdoid phages, etc, but also in the major capsid protein of eukaryotic herpesviruses (*Bamford et al., 2005*). In HK97 itself, the intersubunit contacts are reinforced by a post-assembly covalent linkage—an isopeptide bond between adjacent gp5 subunits, so positioned that the entire capsid is topologically interlinked in a 'chainmail' arrangement (*Duda, 1998*; *Wikoff et al., 2000*). In other cases, such as bacteriophage λ, the non-covalent interactions between the subunits are reinforced by an additional 'cement' protein, which binds on the outer surface of the capsid at positions close to those of the isopeptide bonds in HK97 (*Lander et al., 2008*). Heads of bacteriophage λ defective in this cement protein break down during DNA packaging (*Sternberg and Weisberg, 1977*; *Fuller et al., 2007*; *Lander et al., 2008*).

Bacteriophage BPP-1 is a short-tailed, dsDNA virus and a member of the *Podoviridae* family. It infects and kills *Bordetella* species that cause whooping cough in humans and respiratory diseases in other mammals. It has a T = 7l icosahedral capsid, ~670 Å in diameter. A 7 Å resolution cryoEM structure of BPP-1 showed that its capsid protein has an HK97-like fold, but the shell has an additional protein component (*Dai et al., 2010*). Lacking information at the time about their genetic identities,

**eLife digest** Whooping cough is a respiratory illness caused by bacteria in the Bordetella genus. Among the general public, *Bordetella* species have become a hot topic in recent years due to the re-emergence of whooping cough in the United States and elsewhere. Scientists, meanwhile have become interested in a virus called BPP-1 that can kill the *Bordetella* species.

BPP-1 is a double-stranded DNA virus, and such viruses have long been of interest to scientists because they are the most abundant organisms on Earth. These viruses are also noteworthy because their shells (also known as capsids) are capable of withstanding the very high pressures (up to about 40 atmospheres) that are created by packing so much DNA into the very small volume inside the capsid.

BPP-1 is of particular interest because it is capable of making large-scale changes to its own DNA in order to adapt to changes in its hosts and environment. Of all the organism that do not contain nuclei within their cells (collectively known as prokaryotes), BPP-1 is the only one that is capable of making such changes to its DNA. However, efforts to exploit the properties of BPP-1 for bioengineering applications have been hampered because its detailed structure is not known. Now Zhang et al. have used cryo electron microscopy to study the structure of BPP-1 at the atomic level.

Most viruses belong to one of three major lineages, with each lineage having a characteristic fold in its capsid proteins. Zhang et al. found that BPP-1 contains two of these folds, which suggests that it is a hybrid of two of these lineages. This is the first time that such a structure has been observed. Moreover, Zhang et al. found that one of the folds has an unusual topology that has not been seen before. The atomic structure reveals how double-stranded DNA viruses use a variety of non-covalent interactions and a type of protein 'chainmail' to form a highly stable capsid that is capable of withstanding very high pressures.

In addition to enabling applications in bioengineering, the new structure might also provide insights into the evolution of prokaryotes.

these proteins were named according to their structural roles: a major capsid protein (MCP) and a 'cement protein' (CP) that decorates the shell.

From a biotechnology standpoint, BPP-1 has emerged as an attractive phage-display system thanks to its unique and well characterized diversity-generating retro-element (DGR). As the only known source of massive DNA sequence deviation in prokaryotes, DGRs use a unique reverse transcriptase-based mechanism to introduce targeted diversity into protein-coding DNA sequences to accelerate the evolution of adaptive traits (*Liu et al., 2002*; *Guo et al., 2008*). As such, BPP-1 is a naturally occurring diversity-generating system and can be engineered to display foreign proteins with adaptive heterologous sequences. An atomic description of the BPP-1 head will enhance bioengineering efforts, reveal non-covalent molecular interactions conducive of stable bacteriophage capsid formation, and clarify evolutionary relationships of BPP-1 MCP and CP with proteins in other viruses.

In this study, we report the three-dimensional (3D) structure of the BPP-1 head at ~3.5 Å resolution determined by single-particle cryoEM and derived an atomic model. Both our structural and structure-based mutagenesis studies reveal major novelties in the BPP-1 MCP and CP structures and their interactions. We also show that the C-termini of both MCP and CP are ideally positioned to display DGR-diversified peptide libraries for protein engineering applications.

## Results

### Identification of BPP-1 head proteins by mass spectrometry

Genomic analysis indicated that the 42.5 kb BPP-1 genome encodes up to 50 viral proteins (*Liu et al., 2004*). To identify the genes coding for MCP and CP, we first carried out SDS-PAGE analysis and showed that the two most abundant protein bands have molecular masses of 36.3 kD and 15.2 kD (*Figure 1A*). Mass spectrometry analysis confirmed that the bands correspond to Bbp17 and Bbp16, respectively (*Figure 1B*). The theoretical molecular masses of Bbp17 and Bbp16 (i.e., 36417.2D and 14458.3D) match their apparent molecular masses by SDS-PAGE, suggesting that the two capsid proteins in the mature bacteriophage particles are not cleaved (*Figure 1A*). Finally, to verify that genes

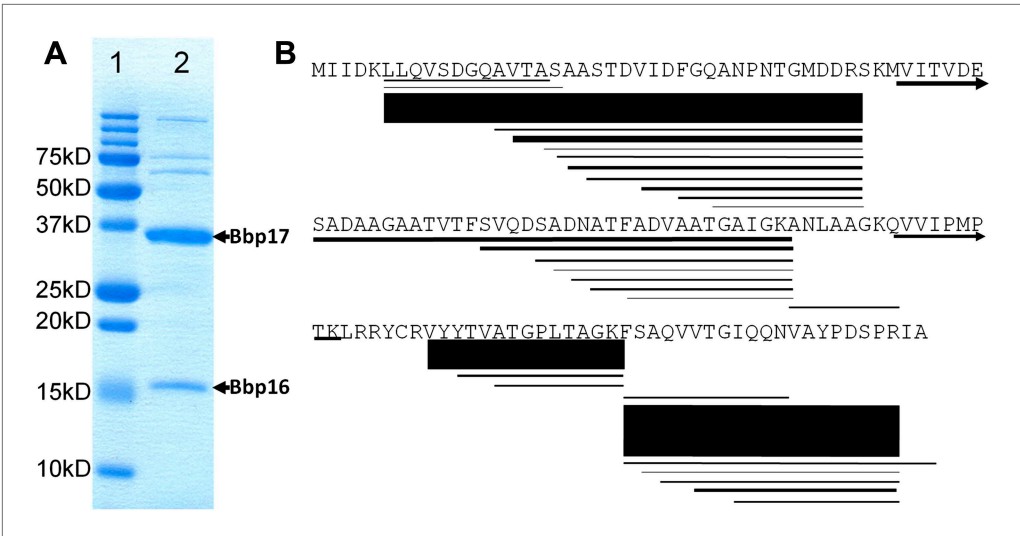

**Figure 1**. Identification of BPP-1 capsid proteins. (**A**) SDS-PAGE of BPP-1 virion proteins stained with Coomassie blue; lane 1: molecular mass standards; lane 2: BPP-1 virion proteins. The two most abundant proteins of the BPP-1 virion are identified by mass spectrometry (**B**) to be Bbp17 (MCP) and Bbp16 (CP) (indicated by arrows). (**B**) Mass spectrometry result of Bbp16 (CP). The sequence is shown with individual peptides identified by mass spectrometry drawn as lines below their corresponding sequences, with line thickness and darkness representing relative abundance in the mass spectrometry profiles (thicker lines mean more abundant). Arrows indicate the two peptide fragments that run past the end of the rows.

*bbp16* and *bbp17* encode components essential for forming infectious phage particles, we constructed in-frame deletions of the *bbp16* and *bbp17* genes. As expected, no infectious phage particles were produced from BPP-1Δ*brt* lysogens carrying deletions in *bbp16* or bbp*17*, confirming that both MCP (Bbp17) and CP (Bbp16) are required for phage production. For the sake of clarity and ease of comparison with structural homologs of other viruses, we will continue to use CP and MCP to refer to Bbp16 and Bbp17 of BPP-1, respectively. As shown below, these assignments are directly verified through atomic modeling in which side-chain structures of amino acid residues in sequences match side-chain densities visualized in the cryoEM map (*Figure 2*).

## CryoEM structure of the BPP-1 head

We reconstructed the 3D structure of the BPP-1 icosahedral head by single-particle cryoEM (*Figure 2*, *Video 1*). The quality of MCP and CP densities was further improved by additional averaging of seven CP or six hexameric MCP subunits (except for the pentameric MCP) in the asymmetric unit (*Zhang et al., 2010a*). Based on the reference-based Fourier shell correlation criterion (FSC = 0.143), the resolution of the capsid is 3.58 Å (*Figure 2C*) (*Rosenthal and Henderson, 2003*). Consistently, the R-factors of atomic models are better than 0.5 at the zone of 1/3.5 Å (for capsid) or 1/3.4 Å (for averaged CP and MCP) (*Figure 2C*; *Table 1*), which correspond to a Fourier shell correlation (FSC) coefficient greater than 0.143 (*Wolf et al., 2010*). Therefore, the resolution of the capsid and averaged densities were estimated to be ~3.5 Å and ~3.4 Å, respectively. This assessment of resolution is also consistent with clearly visible side chain densities of bulky amino acid residues, such as arginine and phenylalanine (*Figures 2D and 4E–F*; *Videos 2–4*).

Arranged on a T = 7 icosahedral shell, MCP and CP each have seven copies or conformers in each asymmetric unit. MCP alone forms a complete icosahedral shell without any noticeable gaps. CP forms dimers bound to the underlying MCP shell at local and icosahedral twofold axes, making the maximum diameter of the BPP-1 (670 Å) bigger than that of the HK97 capsid (659 Å). When CP dimers are computationally removed, the overall size and architecture of the BPP-1 capsid is almost identical to that of the HK97 capsid, and the backbone model of the HK97 capsid is nearly super-imposable with the BPP-1 density map with minor mismatches. Based on the averaged density maps of CP and MCP, we built initial atomic models of CP and MCP with Coot (*Emsley and Cowtan, 2004*), and refined

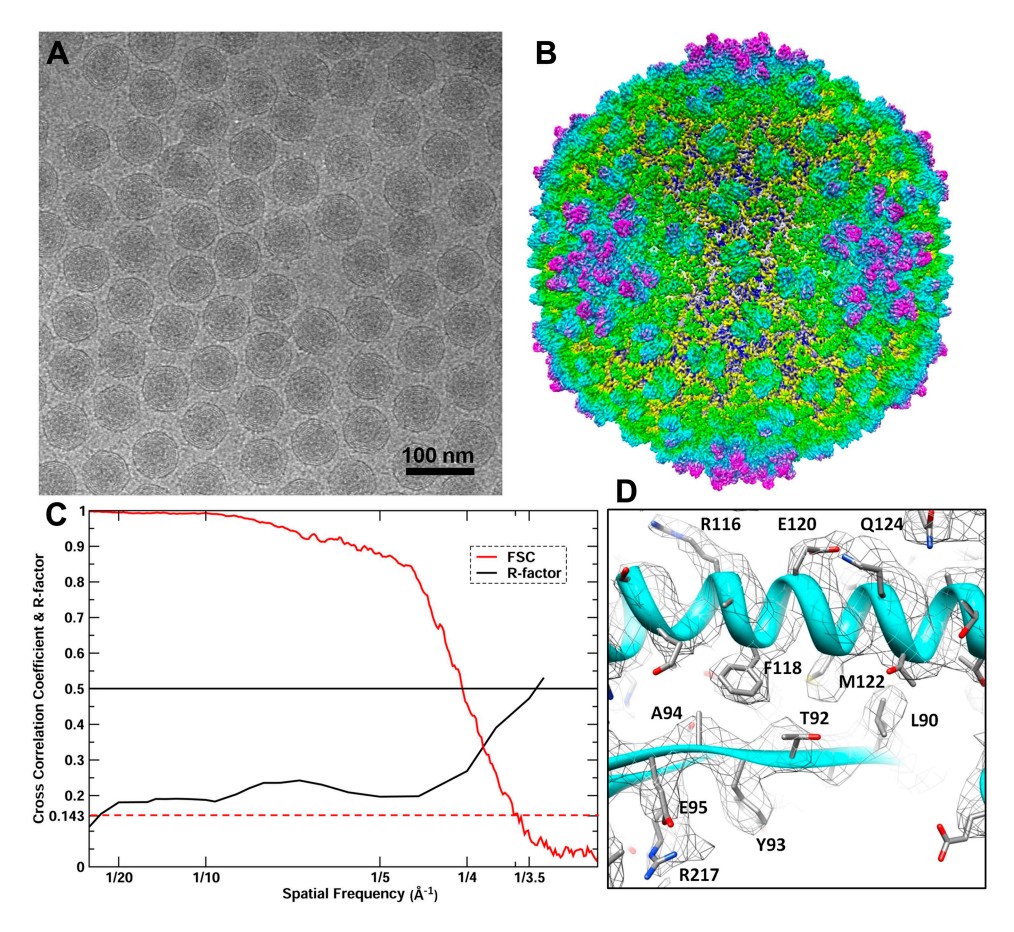

**Figure 2**. CryoEM reconstruction of the BPP-1 head at 3.5 Å resolution. (**A**) Representative cryoEM image (defocus–1.6 µm) of the BPP-1. (**B**) CryoEM density of the BPP-1 head shown as radially-colored surface representation. See also *Video 1*. (**C**) R-factors (red) and Fourier shell correlation coefficients (FSC) (black) as a function of spatial frequency between maps from half datasets. (**D**) Close-up view of a local region of MCP, with densities of many amino acid side chains clearly resolved in both the helix and the loop. The atomic model is shown as ribbons and sticks with amino acid residues labeled.

them with Phenix (*Adams et al., 2010*). Models of the seven MCP and CP monomers within the asymmetric unit were subsequently obtained by adjusting the initial MCP and CP models with Coot and refined with Phenix, with overall R/R-free factors of 0.26/0.27 at 3.5 Å resolution (*Table 1*) ('Materials and methods').

## Cement protein Bbp16 has the jellyroll fold

We traced 139 (Met1 to Ile139) of the 140 amino acid residues of the cement protein Bbp16 in the cryoEM density map. CP contains two β-sheets each consisting of four anti-parallel strands, and a ~36-Å long extension (Thr125 to Ala140) with the C-terminus exposed on the external surface (*Figure 3*; *Video 2*). The backbones of the seven CP subunits in the asymmetric unit, including their C-terminal extensions, are nearly identical, with an RMSD of 0.4–0.66 Å when superimposed (*Figure 3D*).

The topological organization of strands in the two β sheets (i.e., BIDG and CHEF; *Figure 3B*) of CP is identical to that of the jellyroll motif. This structural motif was first discovered in spherical RNA viruses (*Harrison et al., 1978*; *Abad-Zapatero et al., 1980*; *Hogle et al., 1985*; *Rossmann et al., 1985*; *Chelvanayagam et al., 1992*), and subsequently found widely in other DNA viruses, such as φX174, bacteriophage PRD1 and human adenovirus (*McKenna et al., 1992*; *Abrescia et al., 2004*; *Zubieta et al., 2005*; *Liu et al., 2010*; *Krupovic and Bamford, 2011*). Unlike its role as the major capsid protein in the above-mentioned viruses, the jellyroll motif in BPP-1 forms an auxiliary protein to

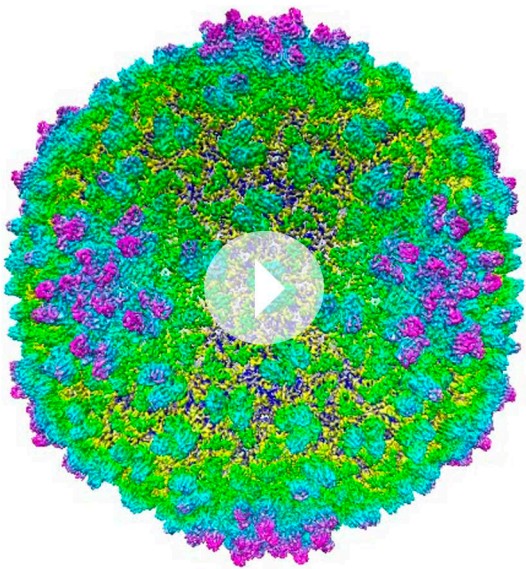

**Video 1**. Shaded surface view of the cryoEM density of the BPP-1 capsid at 3.5 Å resolution. The map is color-coded according to the radius. Related to **Figure 2**.

stabilize the viral capsid made by proteins of another fold. Notably, in some dsRNA viruses, the jellyroll motif has been adopted as a domain in a stabilizer/adaptor protein that forms trimers located at an intermediate layer of viral capsids (**Grimes et al., 1998**; **Mathieu et al., 2001**; **Liemann et al., 2002**; **Zhang et al., 2010a**). In BPP-1, CP subunits form a dimer through β-sheet augmentation between the F-strands of two CHEF sheets, forming an 8 stranded β-sheet visible on the external surface (**Figure 3C**). Although the BIDG sheet of the jellyroll motif faces but do not interact with the underneath MCP shell, instead, as described in detail below, CP interacts with the MCP shell mainly through its N- and C-termini as well as a linking loop between its β-strands B and C.

## Structure of the major capsid protein Bbp17 reveals a new topology of the Johnson fold

We traced 327 amino acid residues (Ser5 to Val331) of the total 331 residues of the major capsid protein Bbp17 in the cryoEM density map (**Figure 4**; **Video 3**). Although little sequence identity (CLUSTALW score = 5) was detected between BPP-1 MCP and HK97 gp5 proteins with ClustalW (**Thompson et al., 1994**), the overall architecture of the MCP resembles that of HK97 gp5 (**Wikoff et al., 2000**) (**Figure 4—figure supplement 1A**), with an axial (A) domain, a peripheral (P) domain and an extended loop (E-loop) (**Figure 4C**). Both the subunit organization and domain orientations of the seven MCPs in the BPP-1 capsid are also identical to those of the corresponding gp5 subunits in the HK97 capsid. The A-domain contains a 5-stranded β sheet flanked by two helices on one side and a C-terminal loop on the other. The P-domain contains a characteristic long, 7-turn helix sandwiched by a three-stranded β sheet and the N-terminal loop. The E-loop contains a single amino acid sequence segment folded into an extended hairpin loop projecting from a 2-stranded β sheet. Notably, both N- and C-termini of MCP are exposed on the external surface of the capsid, thus are accessible for tethering peptides in phage-display applications.

The structures of the seven MCP conformers in the asymmetric unit are nearly identical with some minor differences (**Figure 4—figure supplement 1B**), and RMSDs of their backbones (residues from 31–60 to 81–331) are 0.55–0.98 Å among six hexameric MCPs and 1.2–1.8 Å between

**Table 1.** Statistics of atomic model refinement with Phenix

|  | CP (Bbp16) | MCP (Bbp17) | Asymmetric unit (7CPs+7MCPs) |
|---|---|---|---|
| Residues resolved | 1–140 | 7–331 | 1–140 (CP); 5–331 (MCP) |
| Resolution (Å) | 3.4 | 3.4 | 3.5 |
| $R_{work}$ (overall: 40–3.4 Å) | 0.28 | 0.27 (0.31*) | 0.26 |
| $R_{free}$ (overall: 40–3.4 Å) | 0.28 | 0.28 (0.30*) | 0.26 |
| $R_{work}$ (best resolution zone) | 0.47 (1/3.4 Å) | 0.50 (0.52*) (1/3.4 Å) | 0.51 (1/3.5 Å) |
| $R_{free}$ (best resolution zone) | 0.45 (1/3.4 Å) | 0.45 (0.50*) (1/3.4 Å) | 0.47 (1/3.5 Å) |
| Ramachandran plot values |  |  |  |
| Most favored (%) | 85.5 | 86.7 | 88.0 |
| Generously allowed (%) | 12.3 | 11.8 | 10.3 |
| Disallowed regions (%) | 2.2 | 1.5 | 1.7 |

*R-factors of the interchanged model by forcing the BPP-1 MCP to trace the HK97 topology.

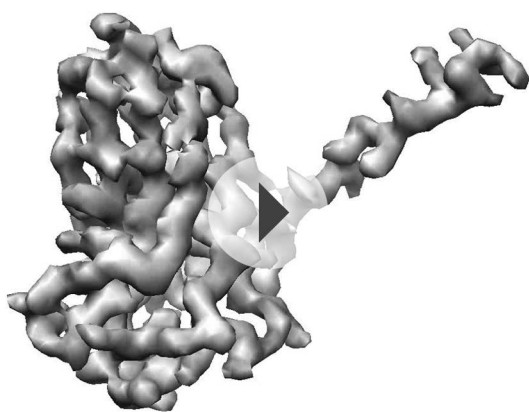

**Video 2**. Shaded surface view of the cryoEM density of the averaged CP at 3.4 Å resolution (gray) superimposed with the atomic model CP (ribbon and sticks). Related to *Figure 3*.

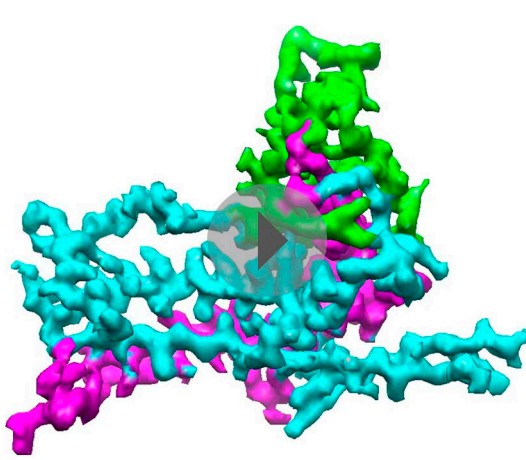

**Video 3**. Shaded surface view of the cryoEM density of the averaged MCP at 3.4 Å resolution. The three structural elements are color coded (cyan, purple and green) according to the structural elements of the Johnson fold as illustrated in *Figure 4*. Related to *Figure 4*.

the pentameric and the hexameric MCPs. The minor differences among the different MCP conformers include: (1) Each N-terminal portion (Ser5–Glu30) of the seven MCPs adopts a slightly different conformation, and backbone RMSDs of this segment are 1.4–5.0 Å among seven MCP copies; (2) The E-loops of the seven MCPs have backbone RMSDs of 0.8–8.3 Å, larger than other portions of MCPs, possibly resulting from different local shell curvatures at the regions of the E-loops (*Figure 4E*); (3) Although the C-terminal proximal loop (Ala295–Val309) of every hexameric MCP is well resolved, the corresponding segment in the pentameric MCP is not resolved, suggesting disordered conformation of this segment in pentons possibly due to steric hindrance at the central channel of the MCP pentamer (*Figure 4—figure supplement 2*).

Despite the similarities in the architectural appearances of BPP-1 MCP and HK97 gp5, their atomic structures differ in four significant ways. Firstly, the conformations of the N-terminal loops are different (*Figure 4—figure supplements 1, 2C*). In BPP-1 MCP, the N-terminal peptide extends radially to interact with a CP by augmenting its CHEF β-sheet. In HK97 gp5, the N-terminus extends circumferentially and interacts with three adjacent gp5 subunits, augmenting a β-sheet in one of them. Secondly, a C-terminal loop (Lys293–Val309) of the pentameric MCP in BPP-1 is disordered while the corresponding loop (Arg294–Thr304) in the HK97 pentameric gp5 is ordered (*Figure 4—figure supplement 2B,C*) (*Wikoff et al., 2000*). Thirdly, the electrostatic potential properties of the two proteins are strikingly different. Finally, and perhaps most interestingly, the folding topologies of BPP-1 MCP and HK97 gp5 are different, which, as described below, provides the first opportunity to explore possible alternative ways to build the highly conserved HK97-like folds (*Figure 5*, *Figure 5—figure supplements 1 and 2*; *Video 4*).

## Two topologies to build the HK97-like ('Johnson') fold and structure-based mutagenesis

As remarked above, despite of the similar architectural appearance (or fold) of BPP-1 Bbp17 and HK97 gp5, the peptide traces of these two proteins follow different folding topologies (*Figure 5A–B*, *Figure 5—figure supplements 1 and 2*). A careful comparison of the two structures has led us to identify three structural elements of the canonical HK97-like, or 'Johnson' fold: N-, β- and α-elements (marked cyan, purple and green in *Figures 4A–D and 5A–C*). These three structural elements join together through a central 5-stranded (F, E, G, K, L in *Figure 5A–C*, in an up-up-down-up-up topology) β-sheet, flanked by two short helices (*Figure 5A–C*, *Figure 5—figure supplement 2*). The β-element (purple) consists of two anti-parallel (i.e., G, K; down-up) β-strands forming a hairpin located at the middle of the Johnson fold. The α-element (green) contains two parallel (i.e., F, E; up-up) β-strands and the two short helices. The N-element (cyan) contributes the last (i.e., L; up) β-strand to the Johnson

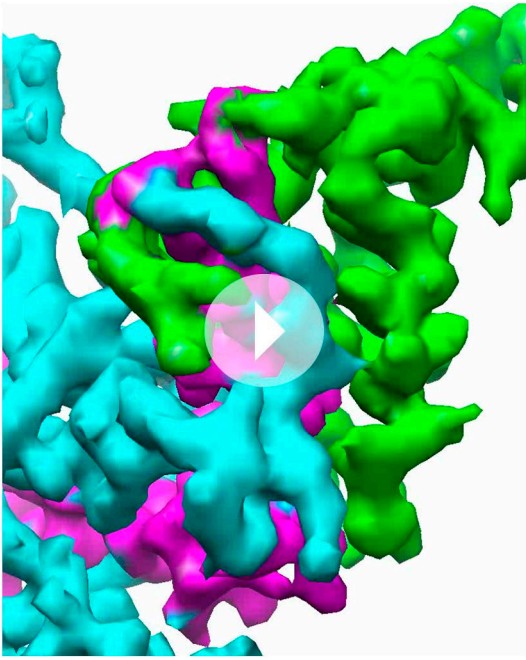

**Video 4**. CryoEM density of MCP around the positions of permutation. The three structural elements are color coded (cyan, purple, and green) according to the structural elements of the Johnson fold as illustrated in *Figure 4*. Related to *Figure 4*.

fold and also contains the two other characteristic secondary structures of the Johnson fold: a long, kinked 'spine' α-helix and the extended loop (E-loop).

In BPP-1 MCP, the N-, β-, and α-elements consist of residues 1–168, 169–241, and 242–331, respectively (*Figure 4A*). The orders of these three structural elements in BPP-1 and HK97 are N-β-α ('BPP topology', *Figures 4A–C and 5A*), N-α-β ('HK97 topology', *Figure 5B*), respectively. To verify this, we swapped the α and β structural elements in our de novo BPP-1 MCP model to create an 'interchanged model' that matches the HK97 topology. This 'interchanged model' was then refined with Phenix for five cycles. Despite this refinement effort, the final 'interchanged model' does not agree with our experimental cryoEM density (*Figure 4—figure supplement 3*). In particular, many side chains in the 'interchanged model' do not match those resolved in the cryoEM density map (*Figure 4—figure supplement 3B,C*). This verification provides additional support to the BPP topology established by our de novo modeling approach.

Interchange of two of the structural elements can lead to proteins with different topologies of the Johnson fold (*Figure 5C*). From a pure mathematical point of view, permuting three structural elements can give rise to a total of six different topologies (i.e., 3! = 6), considering that sequence polarity (N- to C-termini) of the structural elements must be preserved. However, the N-element (cyan) cannot participate in this permutation due to the remote disposition (~42 Å) of its N-terminus with respect to the C-termini of other structural elements, and thus the total possible topologies is reduced to only 2 (i.e., 2! = 2) and the permutation is non-circular (*Figure 5—figure supplements 1A–B, 2G–H*). Remarkably, the newly discovered BPP topology presented here is therefore the second topology predicted from the above rule (*Figure 5A*), the first being the topology discovered in HK97 gp5 (*Wikoff et al., 2000*) (*Figure 5B*) and subsequently seen in many other viruses, as well as in bacteria and archaea cells (*Akita et al., 2007*; *Sutter et al., 2008*). Insertions into any of the three structural elements can give rise to more complex architectures without increasing the number of folding topologies of the Johnson fold, as would be expected for larger viral particles such as the herpesvirus capsid.

To test whether a functional protein can be produced from BPP-1 MCP by permuting it to the HK97 topology without introducing other changes, we genetically interchanged the primary order of the β- and α-elements in the BPP-1 MCP gene *bbp17* (*Figure 5—figure supplement 1C*), and made two slightly different constructs (*Figure 5—figure supplement 1C*) (i.e., PM1 and PM2, 'Materials and methods'). These two engineered proteins were expressed successfully, as indicated by Western blot analysis (*Figure 5D*). However, no plaque formation was detected for the PM1 and PM2 lysates on RB50 cells transformed with either the wt *bbp17*, the PM1 or the PM2 construct (*Figure 5E*), suggesting that the PM1 and PM2 gene products are not functional.

## Interactions among capsid proteins

The BPP-1 capsid has a chainmail structure similar to that of HK97 capsid (*Figure 6A–B*; *Video 5*). Instead of covalent bonding as in HK97, the BPP-1 chainmail is stabilized by non-covalent interactions, such as: (1) strong electrostatic interactions between adjacent MCPs (*Figure 6*), and (2) additional CP-MCP interactions contributed by CPs (*Figure 7*).

Electrostatic interactions between MCPs of BPP-1 are stronger than those between gp5 subunits in the HK97 capsid (*Figure 6D–H*, *Figure 6—figure supplements 1*) for two reasons. Firstly, in the MCP

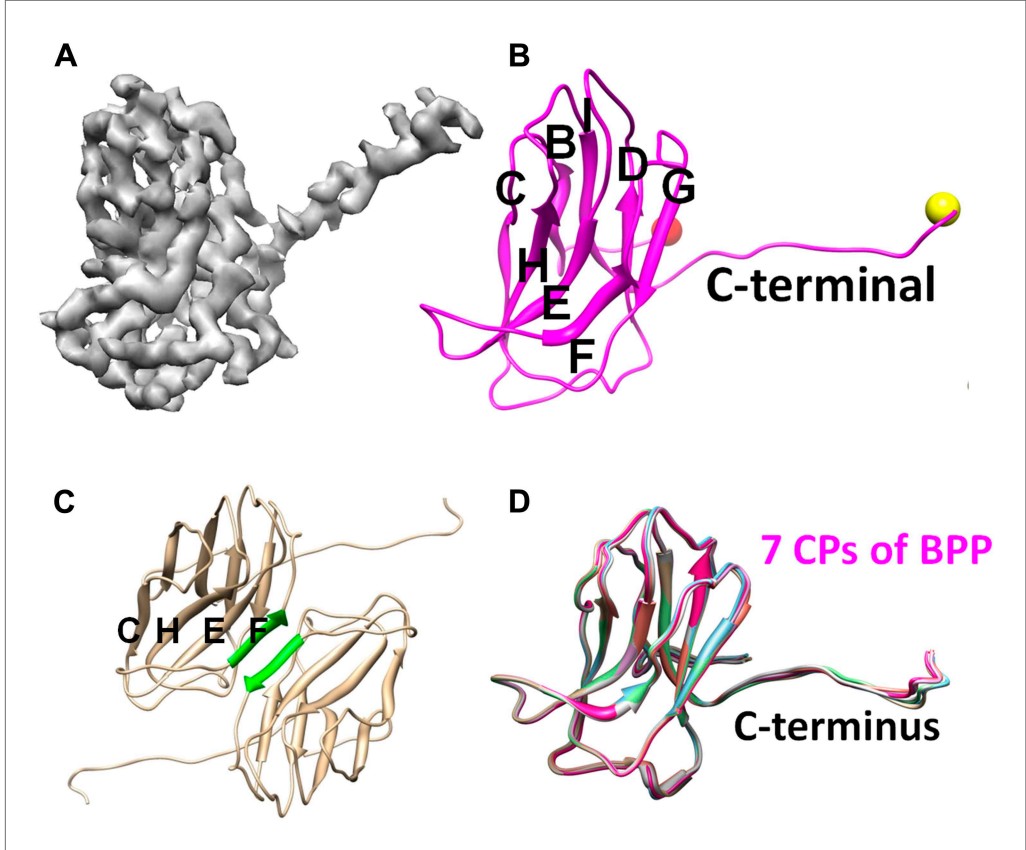

**Figure 3**. Structure of CP. (**A**) CryoEM density map of CP (3.4 Å resolution, average of all seven conformers in an asymmetric unit). See also *Video 2*. (**B**) Ribbon model of CP, showing its jellyroll fold. Eight β-strands (B, C, D, E, F, G, H, I) fold into the two characteristic β-sheets (BIDG and CHEF), forming a 'jellyroll' (*Harrison et al., 1978*; *Abad-Zapatero et al., 1980*; *Hogle et al., 1985*; *Rossmann et al., 1985*). The N- and C-termini are marked by red and yellow balls, respectively. (**C**) The two F strands (green) of two neighboring CP monomers form hydrogen bonds in an antiparallel fashion, creating an augmented, 8-stranded β-sheet and a CP dimer. (**D**) Ribbon diagrams of the atomic models of the seven CP conformers of BPP-1.

hexamer, the electrostatic interactions between adjacent MCPs that appears to be stronger than those between HK97 hexameric gp5 proteins (*Figure 6—figure supplement 1A–B*). Secondly, the interface between the MCP E-loop and an adjacent MCP protein at each local threefold region contains complementary electrostatic interactions (*Figure 6D–E*). Specifically, the E-loop is mainly positive charged and its MCP interface is mainly negative charged, and the interaction at this interface appears to be stronger than that of HK97 gp5 in the same interface. Interestingly, the electrostatic properties of BPP-1 MCP and Hk97 gp5 are opposite in this interface (*Figure 6—figure supplements 1A–B*). The stronger electrostatic interaction between the E-loops and the local threefold region of the BPP-1 MCP may be the reason for the absence (*Figure 8*) of the salt bridge found between HK97 gp5 proteins at the local threefold interface that is critical for the assembly, stability and maturation of the HK97 capsid (*Gertsman et al., 2010*).

In addition, the BPP-1 head is further stabilized by interactions between CP and MCP. Complementary electrostatic potential surfaces were also identified at the interface between CP and MCP (*Figure 7A–C*). Each CP interacts extensively with five MCP subunits underneath, mainly involving three loops: (a) N-terminal loop, (2) C-terminal loop, and (3) the linking loop between strands C and D of the jellyroll motif (*Figure 7D–F*). Firstly, the N-terminal loop of CP interacts with two MCPs: the N-terminal loop of first MCP (green) and the E-loop of the second MCP (yellow) (*Figure 7D–E*), Secondly, the C-terminal loop of CP interacts with two MCPs: a β-strand of the β-element of the first MCP (orange) and both N- and C-terminal loops of a second MCP (green) (*Figure 7D–E*). Thirdly, the

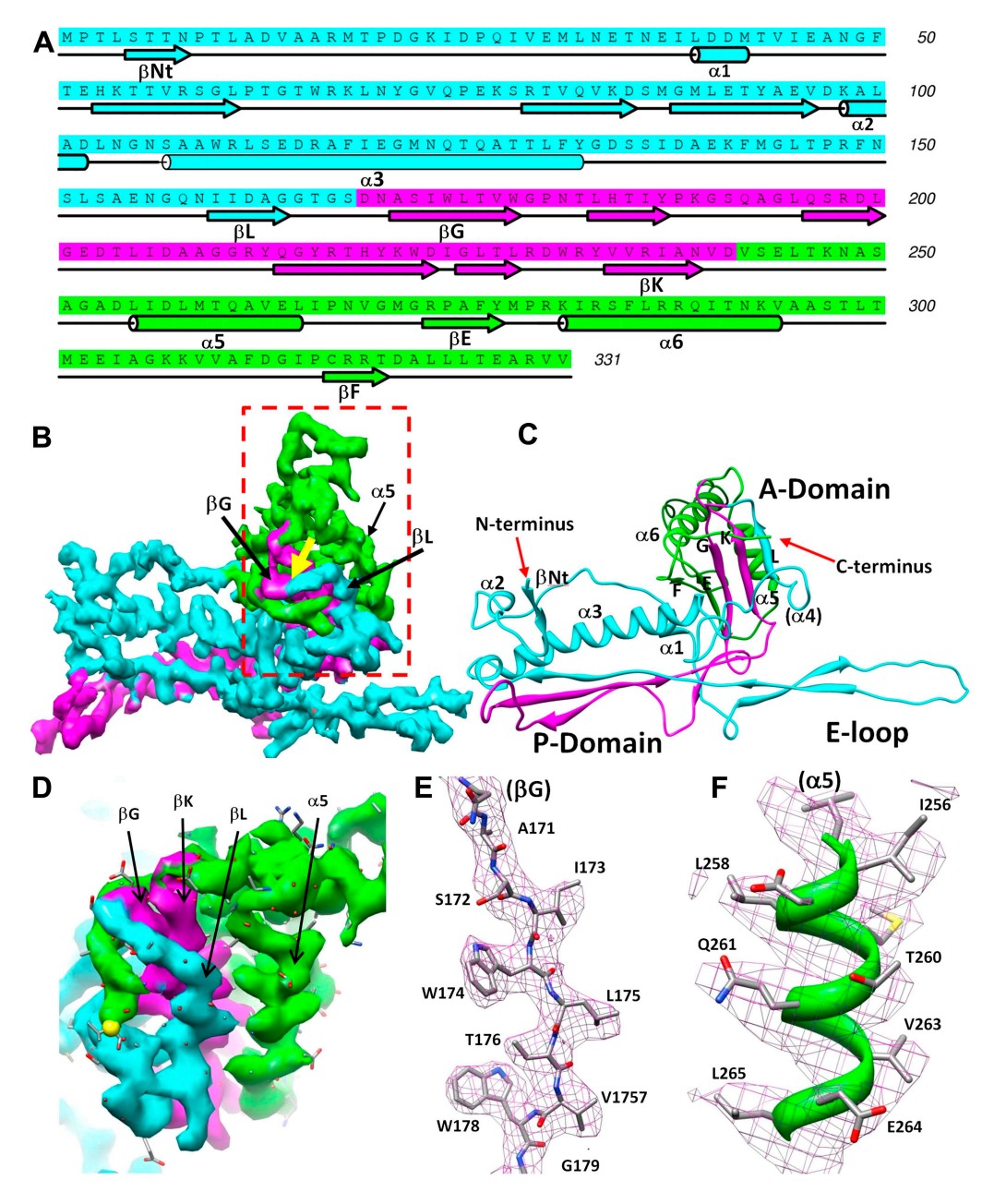

**Figure 4**. Structure of MCP. (**A**) Sequence and secondary structure assignment of MCP. α-helices are marked by cylinders, β-strands by arrows, and loops by thin lines. The three structural elements of the Johnson fold, including N-, β- and α-elements, are shown in cyan, purple and green, respectively. (**B**) CryoEM density map of MCP (3.4 Å resolution, average of the six hexon MCP subunits in an asymmetric unit) shown as shaded surface using the same color coding of (**A**). The dashed box is shown in stereo in (**D**) contains the point of permutation (yellow arrow). See also **Videos 3 and 4**. (**C**) Ribbon diagram of the MCP atomic model with the three structural elements of the Johnson fold colored as in (**A**). (**D**) BPP-1 MCP density within the dash-box drawn in (**B**), showing the positions of the permutation (indicated by a yellow arrow in [**B**]) through which the N-element (cyan) is connected to the β-element (purple), instead of to helix α5 (green, far away), as in HK97 gp5. (**E**) CryoEM density (mesh) of the βG strand in (**D**) superimposed with its atomic model (sticks). (**F**) CryoEM density (mesh) of the α5 helix in (**D**) superimposed with its atomic model (ribbon with sticks).

*Figure 4. Continued on next page*

*Figure 4. Continued*

The following figure supplements are available for figure 4:

**Figure supplement 1**. Structural comparisons between one hexameric MCP of BPP-1 and gp5 of HK97.

**Figure supplement 2**. Differences between BPP-1 pentameric MCP and HK97 gp5.

**Figure supplement 3**. Incorrect MCP model obtained by enforcing the HK97 topology into the BPP-1 MCP cryoEM density, followed by five cycles of model refinement ('interchanged model').

linking loop of CP interacts with three MCPs: two β-strands of the β-element of the first MCP (blue), the N-terminal loop of a second MCP (green) and a short loop (Asn48–Glu52) of the third MCP (orange) (*Figure 7D–E*). Fourthly, β-strand C of CP jellyroll interacts with the N-terminal loop of an MCP, augmenting the CHEF β-sheet of CP (*Figure 9C*). Finally, two CP monomers form a dimer through antiparallel interaction between their F-strands of jellyrolls, forming an augmented, 10-stranded β sheet and further stabilizing the capsid (*Figures 3C, 7D and 9C*).

## Discussion

We found that BPP-1 CP and MCP adopt two of the three characteristic folds of major virus lineages, jellyroll, and HK-97/Johnson. Moreover, the topology of the Johnson fold in BPP-1 MCP is non-circularly permuted as compared to the previously known topology of the Johnson fold. Until now, the Johnson fold has been observed at atomic detail in the *Siphoviridae* and *Myoviridae* families of the three tailed-bacteriophage families, exemplified by gp5 of HK97 (*Siphoviridae*) (*Wikoff et al., 2000*) and gp24 of T4 (*Myoviridae*) (*Fokine et al., 2005*). Our atomic model derived from the 3.5 Å cryoEM map of BPP-1 represents the first atomic model for a virus in the third tailed-bacteriophage family, the *Podoviridae*. Indeed, BPP-1 represents the first among all viruses shown to contain both the jellyroll and HK97-like folds. In the virosphere, the conventional wisdom is that viruses have evolved from a few distinctive lineages separately. Three major lineages have been proposed based on three highly distinctive structures: jellyroll-like (i.e., jellyroll lineage, include single- and double-jellyroll), HK97-like (i.e., HK97 lineage) and BTV-like (i.e., BTV lineage), and members of each lineage contain MCPs with conserved folds originating from a common ancestor (*Rossmann and Johnson, 1989*; *Bamford et al., 2005*; *Abrescia et al., 2012*). Our result suggests that BPP-1 is likely a hybrid virus that has adopted the characteristic structures from both the jellyroll and HK97 lineages.

The head of bacteriophage Epsilon15 also contains two capsid proteins (gp7 and gp10) located at similar positions as CP (Bbp16) and MCP (Bbp17) in BPP-1. In particular, BPP-1 Bbp17 (MCP) shares ~46% sequence identity with gp7 of Epsilon15 phage, indicating that the two proteins are likely to have nearly identical structures at least at the level of protein backbones. Indeed, although initially misinterpreted to have a different topology (*Figure 5—figure supplement 2H–I*) due to limited resolution (*Jiang et al., 2008*), the model of Epsilon15 gp7 was recently updated on the basis of an improved 4.5 Å resolution map (*Baker et al., 2013*) to bear the same BPP topology as we initially reported (*Zhang et al., 2012*) and detailed further in this study.

As described in the results above, the BPP topology and HK97 topology represent the only two feasible topologies of the Johnson fold (*Figure 5*). To date, all other known atomic structures of proteins with the Johnson fold adopt the HK97 topology, including gp5 of HK97 in *Siphoviridae* (*Wikoff et al., 2000*), gp24 of T4 in *Myoviridae* (*Fokine et al., 2005*), a 39kD spherical particle-forming protein in archaea (*Akita et al., 2007*) and an encapsulin protein in bacteria (*Sutter et al., 2008*) (*Figure 5—figure supplement 2A–E*). In these two topologies, the primary positions of their β- and α-elements, are swapped in a non-circularly permutated fashion (*Figure 5A–C*, *Figure 5—figure supplement 1*), which rarely occurs in nature (*Vogel and Morea, 2006*). This, combined with the fact that no recognizable sequence similarity can be identified between the BPP-1 MCP and other HK97-like MCPs (CLUSTALW scores: ~5), obscures the origin of the BPP topology.

From an engineering stand point, it is fortunate that both termini of MCP and the C-terminus of CP are exposed on the external surface of the capsid, ideally positioned for peptide display. Indeed, we observed infectious phage particle formation when either CP or MCP was fused to a peptide of 22 amino acids at their C-terminus (HG and JFM, unpublished observation). As BPP-1 DGRs diversify

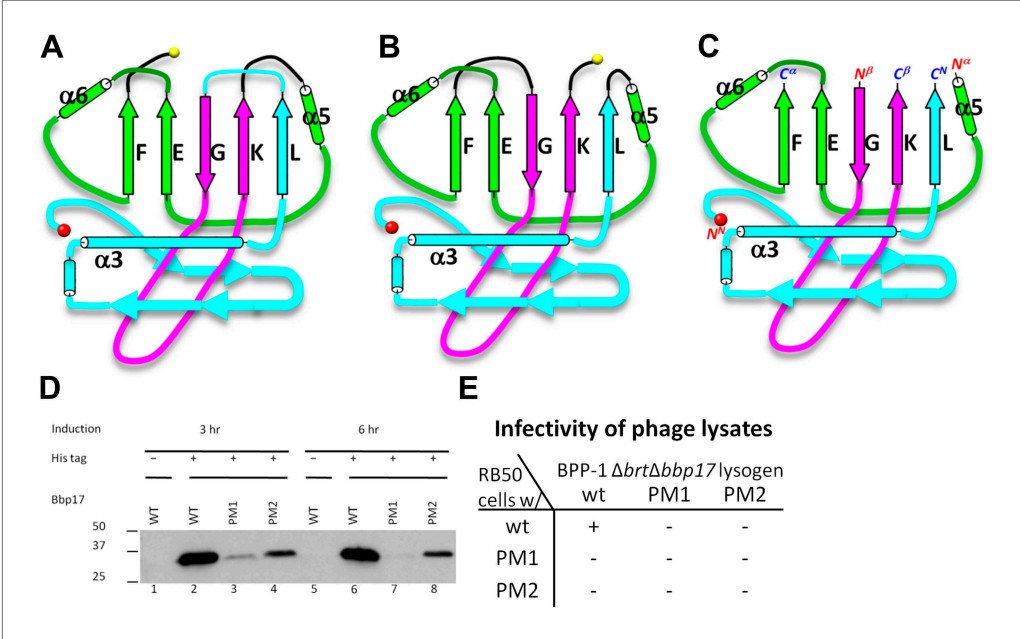

**Figure 5**. Two different topologies of the Johnson fold and structure-based mutagenesis of MCP. (**A**) Diagram of the BPP topology of the Johnson fold: N- , β- , α-elements. (**B**) Diagram of the HK97 topology of the Johnson fold: N- , α- , β-elements. (**C**) Diagram of the three structural elements of the Johnson fold with free N- and C-ends. (**D**) Expression of BPP-1 MCP mutants with the β- and α-elements swapped, thus adopting the HK97 topology. Wild type (WT) and mutant (PM1 and PM2) BPP-1 MCPs with a 6xhistidine tag at the C-terminus were induced for expression from an *fhaB* promoter in RB50 cells for 3 (lanes 1–4) and 6 (lanes 5–8) hr, respectively. The expressions levels were determined by Western blot with a mouse anti-6xhistidine monoclonal antibody. Lanes 1 and 5 are negative controls with wild type *Bbp17* that does not contain a 6xhistidine tag. (**E**) Infectivity of phage lysates as measured by their ability to form plaques on transformed RB50 cells. '+': plaque observed; '−': no plaque observed. The color scheme in (**A**–**C**) is the same as in *Figure 4A–D*.

The following figure supplements are available for figure 5:

**Figure supplement 1**. Diagram of non-circular permutation of the three structural elements of the Johnson fold in BPP-1 MCP and HK97 gp5.

**Figure supplement 2**. Johnson fold in HK97-like proteins.

DNA sequences in vivo, this suggests they may be engineered to display heterologous sequences on the phage capsid by fusion to the termini of CP or MCP. For DGR-based phage-display, the diversity of the DNA library is not limited by the efficiency of DNA transformation, and both the library construction and optimization can occur entirely inside bacterial cells without the need for in vitro manipulation. Moreover, the selected phage particles can be easily re-amplified for iterative selection since the tail, not the engineered head, is involved in phage-receptor recognition. The atomic model of the BPP-1 head presented in this study provides a road map forward to harness the great potentials afforded by the unique properties of BPP-1 for phage-display engineering.

## Materials and methods

### Production and purification of BPP-1

500 ml of LB medium was inoculated with a single colony of BPP-1 lysogen. The cultures were incubated at 37°C on a rotary shaker until log phase when phage production was induced by adding mitomycin C to a final concentration of 2 mg/l. After 3 hr of induction, $CHCl_3$ was added with shaking to facilitate cell lysis and phage release. Cellular debris was removed by centrifugation at 5000×*g*. Phage particles were then precipitated using 10% PEG8000/500 mM NaCl, pelleted by centrifugation, and resuspended in 50 mM Tris-HCl, 250 mM NaCl, 1 mM $MgCl_2$, pH7.5. The resuspended phage

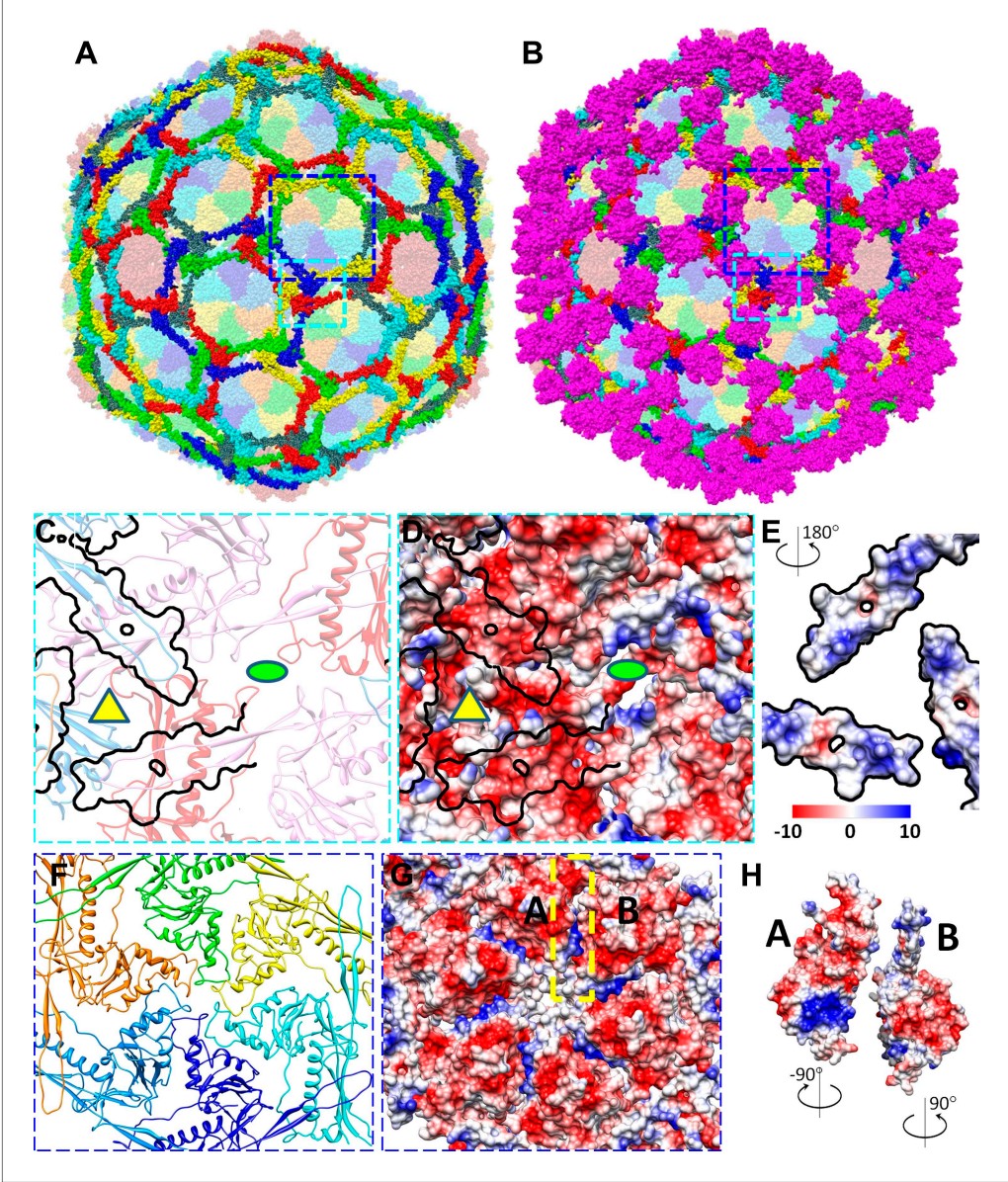

**Figure 6**. Non-covalent chainmail of the BPP-1 head. See also *Video 5*. (**A**) Chainmail network formed by BPP-1 MCPs. The P-domain and E-loop of MCP in neighboring capsomers join head to tail to form rings (bright colors), which concatenate to form non-covalent chainmail. The P-domains and E-loops contributing to the formation of the same ring are shown in the same color. Other domains of MCP are dimmed. (**B**) Same view as in (**A**) but with CP dimers (purple) also shown. (**C–E**) Inter-capsomeric MCP-MCP interactions. The close-up view of all MCPs within the region within the cyan box in (**A**) illustrates the interaction interfaces (outlined by black lines) between the overlying E-loops of three MCP monomers and the underlying domains of other MCPs. These MCP monomers are shown as ribbons in different colors and belong to different hexon (or penton) capsomers. The local threefold and twofold axes are denoted by a yellow triangle and a green ellipse, respectively. Complementary electrostatic potentials are evident at the interaction interfaces, shown separately to reveal the surfaces of the underlying MCPs (**D**) and the 180°-rotated overlying E-loops (**E**), respectively. (**F–H**) Intra-capsomeric MCP–MCP interactions. The region within the blue box of (**A**) contains a hexon with its six MCP monomers shown either as ribbons in different colors (**F**) or as electrostatic potential surfaces (**F**). Adjacent MCP subunits within the hexon share one interaction interface (e.g., the yellow dashed rectangle for subunits A and B in **F**) and their complementary electrostatic potential surfaces are evident in their rotated views (**H**). The electrostatic potential scale is shown in the color bar in (**E**).

*Figure 6. Continued on next page*

*Figure 6. Continued*

The following figure supplements are available for figure 6:

**Figure supplement 1**. Inter-capsomer interactions in HK97 capsid.

particles were further purified by 15–45% (buffered with 50 mM Tris-HCl, 250 mM NaCl, 1 mM $MgCl_2$, pH7.5) sucrose gradient ultracentrifugation at 35,000 rpm for 90 min using an SW41 rotor in a Beckman L8-80M ultracentrifuge. The phage band was visualized by illuminating the gradient tube and carefully collected by side puncture. Purified phage was buffer-exchanged and concentrated in 50 mM Tris-HCl, 250 mM NaCl, 1 mM $MgCl_2$, pH7.5 by ultra-filtration with a MW cutoff of 100 kD (Millipore, Billerica, MA). The final concentration of BPP-1 was ~$10^{14}$ pfu/ml.

## Protein gel electrophoresis and mass spectrometry

To identify capsid proteins using SDS−PAGE and mass spectrometry, purified BPP-1 particles were incubated in 1× SDS sample buffer (0.0625 M Tris-HCl, 1.25% SDS, 5% glycerol and 0.02% bromophenol blue, pH 6.8) at 95°C for 5 min before loading onto the gel. SDS−PAGE was performed using a discontinuous gel (4% polyacrylamide stacking region and 15% polyacrylamide resolving gel) prepared with a discontinuous gel system (Bio-Rad Laboratories, Hercules, CA). Protein bands were visualized by Coomassie Blue staining (Bio-Rad Laboratories). The two major bands (*Figure 1A*) were manually excised, digested in-gel with sequencing-grade modified trypsin (Promega, Madison, WI), and analyzed using Matrix-assisted laser desorption/ionisation-time of flight mass spectrometry (MALDI-TOF MS). Scaffold (version.3.00.04, Proteome Software Inc., Portland, OR) was used to validate MS/MS-based peptide and protein identifications.

## CryoEM imaging, data processing and resolution assessment

To determine the atomic structure, low dose images of liquid-nitrogen-cooled, frozen hydrated BPP-1 were recorded on Kodak SO-163 film on an FEI Titan Krios cryo electron microscope operated at 300 kV with dose of ~25e$^{-1}$/Å$^2$ on specimen. The nominal magnification of images is 59,000×, which was previously calibrated to be 57,660× using tobacco mosaic virus as a standard. Imaging condition was optimized by using parallel illumination and by minimizing beam tilt with a Coma-free alignment procedure.

895 films were recorded and scanned using Nikon Coolscan9000 scanners under step-size of 6.35 μm, corresponding to 1.1 Å/pixel on the specimen scale. Under-defocus value of these films were determined using CTFFIND (*Mindell and Grigorieff, 2003*), and based on the cross correlation coefficients from CTFFIND3, 340 films with defocus values of −0.8~−2.27 μm were selected for further processing. Particles were selected by an in-house automatic procedure and followed by visual screening to remove particles near edge of films, and total 43,156 particles with an image size of 880 × 880 were boxed. Data processing and 3D reconstruction were accomplished with an integrative approach using Frealign (*Grigorieff, 2007*) and eLite3D (*Zhang et al., 2010b*) on high performance computers including graphics processing units (GPU) as previously described (*Zhang et al., 2010a*). For global search, 2× binned images were used to save time. At the end of each cycle of image processing, the effective resolution was

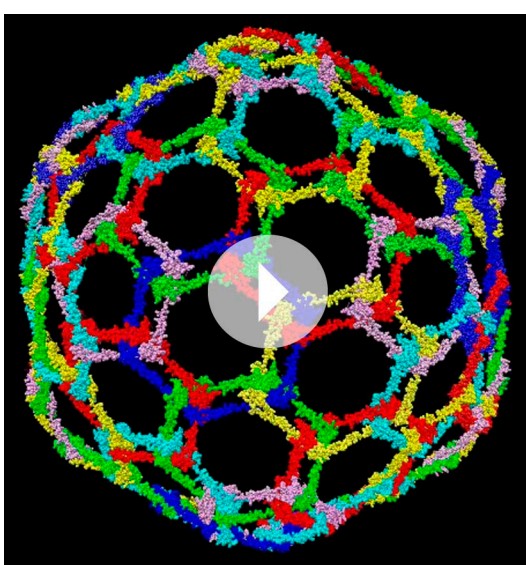

**Video 5**. Non-covalent chainmail formed by MCPs on the BPP-1 capsid. For clarity, only the P domain and the E-loop of MCP subunit are shown. These domains from different capsomers join in a head to tail fashion to form concatenated rings. Ball-and-stick models are produced from our atomic model of BPP-1 and those in the same ring are rendered in the same color. Related to *Figure 6*.

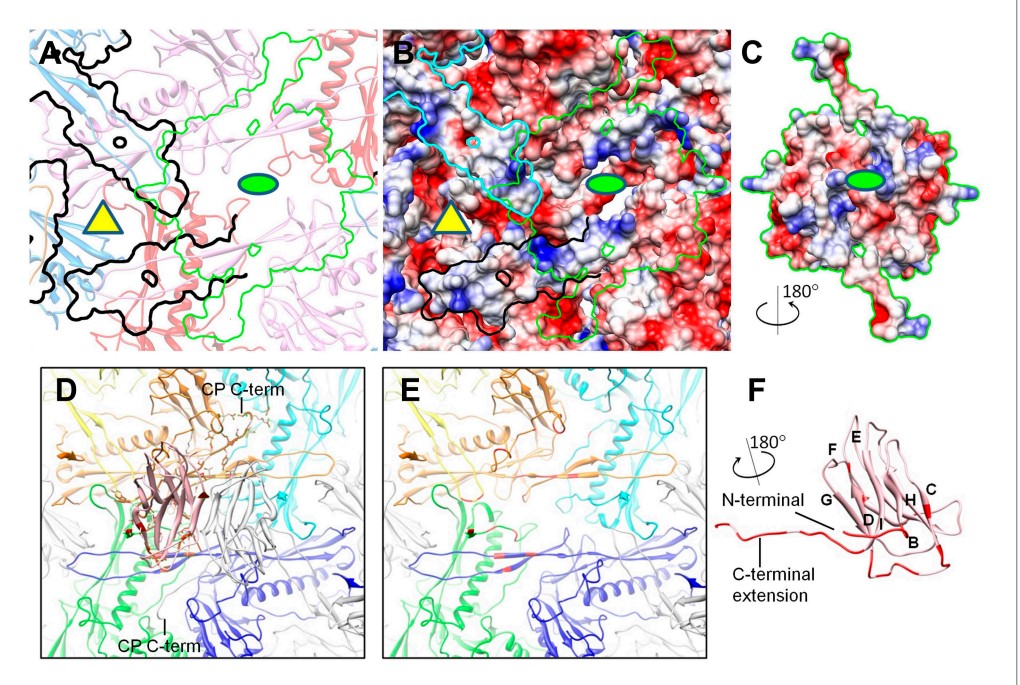

**Figure 7**. Interactions between CP and MCP. (**A**–**C**) The same views as *Figure 6C,D* are shown in (**A**) and (**B**), respectively, except for the addition of a green outline depicting the interaction interface between a CP dimer and its underlying MCPs. At the interaction interface with MCPs, the electrostatic potential surface of the CP dimer (**C**) is complementary to that of its underlying MCPs (**B**). (**D**–**F**) Details of the interactions (segments highlighted in red) between CP and MCP. (**D**) The two monomers in the CP dimer are shown as pink and grey ribbons and the five underlying MCPs as yellow, orange, cyan, blue, and green ribbons. Side chains involved in interactions with the pink CP are shown displayed as sticks. Note the extensive interactions between the CP C-terminal extension and three different MCP monomers. (**E**) The same as (**D**) but without the overlying CP dimer and sticks of interacting side chains to better reveal MCP segments involved in the interactions (red). (**F**) The bottom view of the pink CP monomer to better reveal its segments (red) involved in CP–MCP interactions, which include its N-terminal loop, C-terminal loop, and the loop connecting strands C and D in the jellyroll.

determined and refinement for the next cycle would be carried out by including data up to this resolution. In the last refinement cycle, data up to a spatial frequency of 1/3.7 Å$^{-1}$ was included for the refinement and 39,549 particles were used for the final reconstruction. To further improve signal/noise ratio, local averaging of seven CP or six hexameric MCP subunits (the pentameric MCP is slightly different from the hexameric MCP subunits and was not included in the averaging) in the asymmetric unit was performed as previously described (*Zhang et al., 2010a*).

The effective resolutions were estimated based on the 0.5 R-factors between the cryoEM density map [equivalent to FSC≥0.143 (*Rosenthal and Henderson, 2003*; *Wolf et al., 2010*) and the final atomic model calculated by Phenix (*Adams et al., 2010*). The calculated R-factors reached ~0.5 at 3.5 Å, 3.4 Å for the capsid density and the averaged CP and MCP densities, respectively (*Table 1*). These estimations are consistent with the structural features present in the maps (*Figure 2*, *Videos 2–4*). The capsid and averaged maps were filtered to 1/(3.4 Å) and 1/(3.5 Å) spatial frequency, respectively; and sharpened using a reverse B-factor of −200 Å$^2$ (for capsid) or −250 Å$^2$ (for averaged densities), which were estimated through a trial-and-error procedure by optimizing side-chain densities and noise level simultaneously. Visualization and segmentation of density maps were done with UCSF Chimera (*Pettersen et al., 2004*).

## Atomic modeling and model refinement

Based on the averaged density maps of CP and MCP, we first built initial C$_\alpha$ and full atom models for CP and MCP with Coot (*Emsley and Cowtan, 2004*) without referring to any existing models of other proteins. For MCP, the N- and C-termini were distinguished based on the 'Christmas tree'

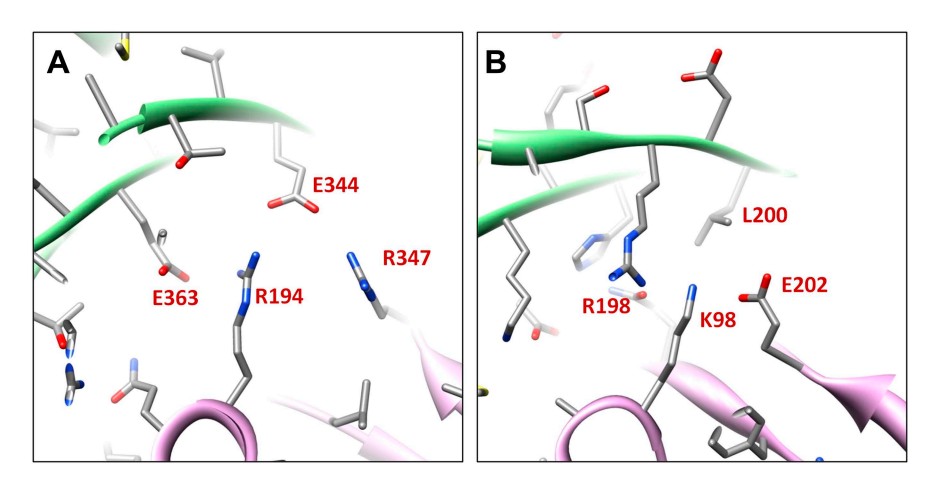

**Figure 8**. Salt bridges between adjacent HK97 gp5 molecules and lack of them in the corresponding positions in BPP-1 MCP molecules. (**A**) Close-up view of two adjacent HK97 gp5 subunits, showing salt bridges between Arg194, Arg347 and Glu363 and Glu344. (**B**) There are no such salt bridges in the corresponding regions in the BPP-1 MCP molecules.

polarity of α helices and confirmed by landmark, bulky side chain densities. For CP, because there is no helix to be used to reveal the N- to C-terminal polarity, we determined the N-terminus using the side chain densities of some landmark amino acids, such as Phe27, Tyr 102, Tyr106, Tyr107 and Tyr133. The initial full atom models were regularized by constraining both Ramachandran geometry and secondary structures in Coot (**Emsley and Cowtan, 2004**) but without including hydrogen atoms.

These initial full atom models were iteratively refined using structural information of both amplitude and phase (from Fourier transform of the cryoEM maps) in the following three distinct (two automatic and one manual) steps. The first step (automatic) was performed in Phenix (**Adams et al., 2010**) using Ramachandran restrain. The second step (automatic) is to regularize the new model also using Phenix.

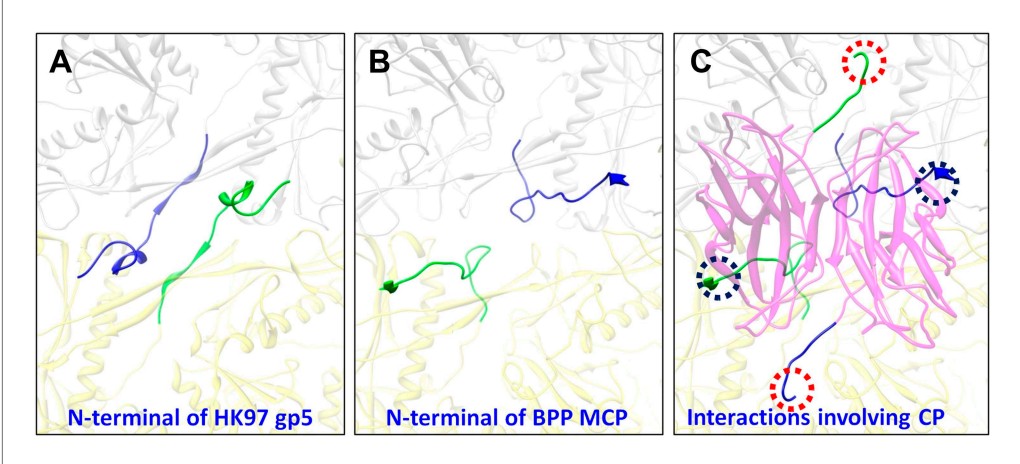

**Figure 9**. Interactions between CP and MCPs. (**A**) Ribbon models of two adjacent HK97 gp5 subunits, showing the interaction of the N-terminal loop (bright colored) with an adjacent gp5 (dimmed). (**B**) BPP-1 regions corresponding to that in (**A**) showing the lack of the HK97 type of interactions between the MCP N-terminal loop (bright colored) and adjacent MCP molecules (dimmed). (**C**) The same region in (**B**) but with the CP dimer (purple), showing that the N-terminal loop of MCP hydrogen-bonds to β-strand C of CP (black circles) to form an augmented 10-stranded β sheet, and that the C-terminal loop (red circles) of each CP has extensive interactions with nearby MCP molecules.

To do so, hydrogens were added to all atoms of the model from the last refinement, and followed by regularization and removal of hydrogens. The latest models were refined iteratively until no further improvement was apparent based on both Ramachandran geometry and R-factors. Then in the third, manual step, amino acid residues with invalid Ramachandran backbone geometries were identified and their backbone Psi–Phi angels were manually corrected in Coot. This process of automatic- and manual model refinement steps was iterated until no further improvement on both Ramachandran geometry and R-factors was evident.

Atomic model of an asymmetric unit including seven MCPs and seven CPs was subsequently obtained by adjusting the refined MCP and CP models according to the density maps of different conformers. This model of the asymmetric unit was refined with Phenix under the constraints of Ramachandran geometry, secondary structures, and icosahedral symmetry (*Adams et al., 2010*). Clashes at the molecular boundaries across different asymmetric units in the entire capsid were minimized by including icosahedral symmetry constraints in this refinement. The final atomic model of the full capsid was obtained after 14 cycles of refinement.

Because our BPP-1 MCP chain trace differs from that of the HK97 gp5 in the order of the α and β structural elements ('Results'), we made an extra cautious effort to verify our trace by swapping the α and β structural elements in our model to create an interchanged model that matches the HK97 topology. This interchanged model was then refined with Phenix (*Adams et al., 2010*) for five cycles. Most of the side chains of the refined interchanged model do not match those in the cryoEM density, further confirming our de novo model (*Figure 4—figure supplement 3*).

## Structure-based mutagenesis

To engineer BPP-1 MCP to match the topology of HK97 gp5, we swapped the order of the β- and α-elements in BPP-1 MCP (*Figure 5—figure supplement 1C*). In construct PM1, the β-element (peptide 169–241) was cut from wt *bbp17* gene and then pasted to the C-terminal end of α-element. Construct PM2 was obtained the same way except that the cut sites were shifted by three amino acid residues on both sides, resulting in a six residue-longer β-element (peptide 166–244). In both the constructs, the N-terminal end of the α-element was pasted to the C-terminal end of the N-element.

Plasmids expressing either wt bbp17, PM1 or PM2 genes *B. bronchiseptica* were transformed into RB50 cells transformed by electroporation. These transformed RB50 cells were grown on Bordet–Gengou agar containing 15% sheep blood, 25 µg/ml streptomycin and 25 µg/ml chloramphenicol. The protein expressing levels of the wt *bbp17*, *PM1* and *PM2* genes (each tagged with 6xhistidine at the C-terminus) in these RB50 cells were determined by Western blot. Single colonies were inoculated into Luria–Bertani media containing 25 µg/ml streptomycin, 25 µg/ml chloramphenicol and 10 mM nicotinic acid and grown at 37°C overnight. Cells in the amount of 1 ml at $OD_{600}$ = 1.0 were pelleted and then grown in 2.5 ml of Stainer Scholte media with 25 µg/ml streptomycin and 25 µg/ml chloramphenicol to induce expression of the *bbp17*, *PM1* or *PM2* gene. Cells expressing each construct were grown for 3 and 6 hr. Equal amounts of the cells (by $OD_{600}$) were harvested and lysed by boiling in 1 × SDS-PAGE loading buffer, and subsequently analyzed with SDS-PAGE. Western blot was done with a mouse monoclonal antibody against 6xhistidine as the primary antibody and a horse reddish peroxidase-conjugated goat anti-mouse antibody as the secondary antibody with an Amersham kit.

To obtain phage lysates for plaque assays, we generated a lysogen (BPP-1Δ*brt*Δ*bbp17*) that has the *bbp17* gene deleted. Then, we transformed plasmids expressing either the *wt* bbp17 gene, the PM1 or PM2 genes into the BPP-1Δ*brt*Δ*bbp17* lysogens. These lysogen cells were first grown at 37°C for 3 hr in Stainer Scholte medium containing 25 µg/ml streptomycin and 25 µg/ml chloramphenicol to induce to the Bvg+ phase, leading to the expression of the wt *bbp17* and the two mutants PM1 and PM2. Mitomycin C (0.2 µg/ml) was then added to the cell cultures to induce phage production. After 3 hr, chloroform was added to the cultures, followed by vortexing and centrifugation to remove cellular debris. The resultant supernatants were collected for plaque assays on *B. bronchiseptica* RB50 cells transformed with plasmids expressing either wt bbp17, PM1 or PM2 gene. As above, these transformed RB50 cells were grown on Bordet–Gengou agar containing 15% sheep blood, 25 µg/ml streptomycin and 25 µg/ml chloramphenicol.

## Visualization

CryoEM density maps, atomic models and surface charge properties were visualized with Chimera (*Pettersen et al., 2004*).

## Accession numbers

The cryoEM density map and the atomic model of BPP-1 have been deposited to databanks with accession numbers EMD (5764, 5765, 5766) and PDB (3J4U), respectively.

## Acknowledgements

We thank Pavel Afonine and Paul Adams for advice on using Phenix to refine the atomic models. We acknowledge the use of the cryoEM facility in the Electron Imaging Center for Nanomachines by NIH (1S10RR23057 to ZHZ) and CNSI at UCLA.

## Additional information

### Competing interests

JFM: Jeff F Miller is a cofounder, equity holder, and chair of the scientific advisory board of AvidBiotics Inc., a biotherapeutics company in San Francisco. The other authors declare that no competing interests exist.

### Funding

| Funder | Grant reference number | Author |
| --- | --- | --- |
| National Institutes of Health | AI046420, GM071940 | Z Hong Zhou |
| National Institutes of Health | AI069838 | Jeff F Miller |

The funders had no role in study design, data collection and interpretation, or the decision to submit the work for publication.

### Author contributions

XZ, LJ, EC, AH, Acquisition of data, Analysis and interpretation of data, Drafting or revising the article; HG, Acquisition of data, Analysis and interpretation of data, Drafting or revising the article, Contributed unpublished essential data or reagents; WHH, AWN, Acquisition of data, Drafting or revising the article; JFM, ZHZ, Conception and design, Analysis and interpretation of data, Drafting or revising the article

## Additional files

### Major datasets

The following datasets were generated:

| Author(s) | Year | Dataset title | Dataset ID and/or URL | Database, license, and accessibility information |
| --- | --- | --- | --- | --- |
| Zhang X, Guo H, Jin L, Czornyj E, Hodes A, Hui WH, Nieh AW, Miller JF, Zhou ZH | 2013 | A new topology of the HK97-like fold revealed in *Bordetella* bacteriophage: non-covalent chainmail secured by jellyrolls | 3J4U; http://www.pdb.org/pdb/search/structidSearch.do?structureId=3J4U | Publicly available at the Protein Data Bank (http://www.rcsb.org/pdb/). |
| Zhang X, Guo H, Jin L, Czornyj E, Hodes A, Hui WH, Nieh AW, Miller JF, Zhou ZH | 2013 | A new topology of the HK97-like fold revealed in *Bordetella* bacteriophage by cryoEM at 3.5A resolution | EMD-5764; http://www.ebi.ac.uk/pdbe/entry/EMD-5764 | Publicly available at the Electron Microscopy Data Bank (http://http://www.ebi.ac.uk/pdbe/emdb/). |
| Zhang X, Guo H, Jin L, Czornyj E, Hodes A, Hui WH, Nieh AW, Miller JF, Zhou ZH | 2013 | A new topology of the HK97-like fold revealed in *Bordetella* bacteriophage by cryoEM at 3.5A resolution | EMD-5765; http://www.ebi.ac.uk/pdbe/entry/EMD-5765 | Publicly available at the Electron Microscopy Data Bank (http://http://www.ebi.ac.uk/pdbe/emdb/). |
| Zhang X, Guo H, Jin L, Czornyj E, Hodes A, Hui WH, Nieh AW, Miller JF, Zhou ZH | 2013 | A new topology of the HK97-like fold revealed in *Bordetella* bacteriophage by cryoEM at 3.5A resolution | EMD-5766; http://www.ebi.ac.uk/pdbe/entry/EMD-5766 | Publicly available at the Electron Microscopy Data Bank (http://http://www.ebi.ac.uk/pdbe/emdb/). |

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
