## [Decision Letter]

Thank you for sending your work entitled “A new topology of the HK97-like fold revealed in *Bordetella*bacteriophage: non-covalent chainmail secured by jellyrolls” for consideration at *eLife*. Your article has been favorably evaluated by a Senior editor and 3 reviewers, two of whom, Stephen Harrison and NikoGrigorieff, have agreed to reveal their identity.

Here is a combined summary of the reviews, with the five key areas that need attention before the paper can be accepted:

The structure of the *Bordetella* phage BPP-1, determined by cryoEM at a resolution of about 3.5 Å, contains three broadly interesting findings. The first of these is a description of the major capsid protein chain trace, showing an interchange (in the sequence) of structural elements, with respect to their order HK97. The second is the structure of the “cement” protein, which is present in a number of other bacteriophages (e.g., lambda) for which chain-trace resolution structures have not been available. The third is the overall structure of the capsid, and information about the positions of the N- and C-termini of the two proteins, because of the potential application to a phage display platform. A revision that convinces us even more completely of the chain trace and that pays attention to several additional issues should be appropriate for publication in *eLife*.

Chain trace: the difference between the chain trace for this virus (which is the same as the “revised” trace from Houston for phage epsilon15) and HK97 is curious, although the authors (whose work is, as usual for this group, technically superb) present good evidence for it. Some additional tests and illustrations are requested in time (4) below. Because the Houston group’s trace of epsilon15 was initially wrong (they've now revised it to a version that agrees with the BPP-1 trace), the present authors need to be completely convincing. (The authors have to convince us completely that they now have it right, even if HK97 and T4 and other viruses have it the other way.)

The following revisions should make the paper suitable for publication. Overall, it is too long and redundant, and the illustrations are often unclear. In addition to avoiding repetition (they can easily shorten the paper to no more than 2/3 its current length, even with the additions suggested below), the authors need to pay attention to the following five points.

1) The authors misunderstand and misstate the history of discovery and understanding of viral capsid subunit structure.

2) They need to provide more details about data processing.

3) They confuse structural observations with energetic ones, so their speculations about “withstanding high pressures” (moreover, is 40 atm really that high, on a molecular scale?) are misplaced.

4) They need to present some further validation of the difference between BPP-1 and HK97 and to describe the chain trace and the overall structure more clearly. The current description makes it impossible for anyone except the initiated to understand the interesting relationship between BPP-1 MCP and the HK97 coat protein, unless the reader realizes that all they need do is look at Figure 6. Moreover, the description of structural “blocks” is misleading, as the two strands in the central sheet and the long beta hairpin are parts of distinct structural modules.

5) They should tone down evolutionary speculations. (Francis Crick once said something to the effect that we should always imagine evolutionary scenarios and talk about them, to sharpen our thinking, but we shouldn't publish them, unless there is good evidence.)

1) History. It was never assumed that “all viruses would use the jellyroll fold”. There were, of course, lumpers and splitters – at least one reviewer was basically agnostic (see Crick quote). But the RNA phage structure (1990) showed right away that you could make a perfectly nice, T=3 capsid with a different fold. Amusingly enough, that fold has only been found in RNA phage. Anyhow, the dsRNA viruses and HK97 show two more designs, although most of the dsRNA viruses have another protein (e.g., VP6 of rotavirus) that does have a jellyroll. So while it remains puzzling why jellyrolls show up in so many non-enveloped viruses, playing the “my fold is more curious/novel/interesting than your fold” game is silly.

To rectify the misunderstanding, and to avoid unnecessary “fold-ology” (folds are hardly the most interesting thing about viruses), the following two paragraphs are a proposed revision and condensation of the relevant sections. The revision takes into account two points from item (3) – the misunderstanding about pressure and the proper use of the “chainmail” analogy.

“Capsid proteins of non-enveloped viruses fall, so far, into three major structural classes: the β-jellyroll, the HK97 fold, and the fold of dsRNA-virus shell proteins. The RNA phage subunits have a fourth structure, not yet found in eukaryotic viruses. The HK97 fold is present not only in a large number of dsDNA bacteriophage capsids, including the T-phages, lambdoid phages, etc., but also in the major capsid protein of eukaryotic herpesviruses. In HK97 itself, the intersubunit contacts are reinforced by a post-assembly covalent linkage - an isopeptide bond between adjacent gp5 subunits, so positioned that the entire capsid is topologically interlinked in a “chainmail” arrangement. In other cases, such as lambda, the non-covalent interactions between the subunits are reinforced by an additional “cement” protein, which binds on the outer surface of the capsid at positions close to those of the isopeptide bonds in HK97. Heads of lambda phage defective in this cement protein break down during DNA packaging.”

“BPP-2 is a short-tailed, dsDNA virus, a member of the Podoviridae family. It infects and kills *Bordetella* species that cause whooping cough in humans and respiratory diseases in other mammals. It has a T=7 icsoaehedral capsid, ∼670 Å in diameter. A 7 Å resolution cryoEM structure of BPP-1 showed that its capsid protein has an HK97-like fold, but further suggested that the shell has an additional protein component. Lacking information at the time about their genetic identities, these proteins were named according to their structural roles: a major capsid protein (MCP) and a “cement protein” (CP) that decorates the shell.”

2) Details about data processing. The authors should include, in the section of “CryoEM image, data processing and resolution assessment”, answers to the following questions: i) How was the magnification calibrated? ii) At what resolution was the refinement done? iii) Was pixel binning used? iv) What filtering of the final map was done (low-pass, B-factor)? v) How many images were collected? vi) How many particles were selected? vii) How many particles were used in the final reconstruction? viii) How were “good” particles selected? ix) How was the CTF determined? x) What defocus range was used? xi) How big a box was used for particle windowing?

One reviewer was not particularly impressed by an R-factor comparison between calculated and observed maps, especially if 0.5 is the cut-off. Refinement and free R do not mean quite the same thing here as in x-ray refinements. The structure factors against which one refines are already correlated, so saving out a test set doesn't mean much, and the non-capsid symmetry (ncs) averaging correlates them further. Visual comparison of the maps is obviously the most useful criterion, and that is indeed the criterion on which the authors have largely relied. Use of the Scheres“gold standard” might be helpful to assess the resolution more correctly. Even if the authors choose not to do that calculation, full FSC and R-factor curves should be shown because their shapes sometimes indicate problems with the refinement, e.g., over-fitting.

3) Structure and energetics. It turns out that 40atm is not so difficult a pressure to withstand. 40 atm is 4 MPa. To expand the shell by ΔR = 1 Å, probably a reasonably tolerable range for the intermolecular forces holding the shell together (the circumference would change by 6 Å, distributed over more than six intersubunit contacts as you go around, so any intersubunit interface would stretch by less than 1Å, even assuming that the subunits themselves were utterly inextensible), the DNA would do work PΔV ∼ 4 x 10^-18 Joules or 2400 kJ/mole (taking 300 Å as the radius). That seems a lot, but it is only about 5.5 kJ/mole of subunit (there are 420), or roughly the strength of an extra H-bond (i.e., 1.5 kcal/mole). Not particularly impressive. So the chainmail of HK97 is probably not about withstanding “enormous forces”, but rather about withstanding the slings and arrows of the slime through which the poor phage particle must pass to get from one bacterial host to another. That doesn't make the alternative strategies of cement protein and isopeptide bond less interesting – but it is important to avoid hyperbole based on intuitive macroscopic analogies.

“Chainmail” requires topological interlinks. You can't have “noncovalent chainmail”, without stretching the armor analogy farther than a reasonable metaphor should go.

4) Chain trace comparisons.

First, it is essential to be 100% sure that the chain trace is correct. The authors should show clear examples of how an alternative trace violates the density. In particular, as the altered connections are at the surface of the particle, where local disorder, etc., could obscure correct links, they should show that any plausible sequence register for the HK97-like trace is not possible. One simple way to do this would be to examine a series of possible alternative sequence registers for residues following the first of the “non-HK97-like” connections and show that the pattern of hydrophilic and hydrophobic residues is completely different. In other words, that if you look at the possible sequence registers for the long beta-hairpin that runs parallel to the long helix (probably just 4-5 alternative starting and ending points), do any of them look plausible enough to test by the refinement criteria that Zhang et al. use for their trace? A suitable figure showing various sequence registers might be all that is needed to make this point.

The authors should also show density, as in Figure 2, for two segments that are affected by the interchange, to show that there is no ambiguity in how the interchanged sequences fit density. Density extending from the new junctions and including side chains consistent with the sequence in the reordered form would be the best way to do this.

Second, assuming the trace to be correct, the description is confusing and doesn't follow the way we generally describe domains or modules in proteins. First, there are only two modules here that can reasonably be called “domains” (although this is a funny protein, so I'm not sure that “domain” is the right word). The “blocks” that the authors name here are not proper folding units, as the long β-hairpin in their β-block is really part of the P-domain (to use the terminology in Figure 4) and the β-strands are part of the A-domain. Thus, Figure 6 are misleading. They are formally correct, but they do no correspond to how we imagine a protein might fold. The authors should stick to FigureA and C and not try to force the question with three “building blocks” as in Figure 6, the unnecessary calculation of possible “permutations”, etc.

Third, the right word for what has happened here is an “interchange” of the order in the polypeptide chain of a long, two part β-element (effectively a β-hairpin with a right-angle bend, which allows it to participate in two distinct sheets). This element is at the C-terminus in HK97, and it has been pasted in front of strand E (L is barely a strand in HK97) in BPP-1. “Non-circular permutation”, while perhaps a mathematically correct use of the word “permutation”, will confuse phage biologists, for whom “circular permutation” has an important reference in the way T-even phage genomes relate from particle to particle.

5) Evolution and structure comparisons. We have no idea how viral evolution relates to host organism, species jumping, etc. It is not a great idea to speculate about “popular views” of how viruses “originated”. While anyone who contributes solid data to the published literature buys a license for a few paragraphs of speculation, and the authors are entitled, should they wish, in the paragraph beginning “One popular view argues that viruses originated from ancient cells…”, it might be wise to reduce or eliminate it.

Three other points in connection with structure comparisons.i) Is the hand laevo or dextro? All T=7 capsids examined so far are laevo; if BPP-1 turned out to be dextro, this would be another distinctive feature (what is epsilon 15?). ii) When considering subunit folds of different lineages, many authors differentiate between the single jellyroll fold observed in the first ssRNA virus structures determined and the double jellyroll found in dsDNA viruses infecting all three domains of life. When multiple jellyrolls from RNA viruses are in a single polypeptide chain like the large subunit of cowpea mosaic virus or all 3 subunits of tobacco ring spot virus they clearly appear as “beads on string”, while the double jelly roll structure integrates the two domains into a contiguous protein. iii) The authors state that BPP1 probably has a different assembly and maturation pathway compared to HK97. In order to make this statement meaningful they need to discuss the scaffolding protein of BPP1 and any intermediates in maturation that may have been characterized. HK97 has a covalently associated scaffold that is proteolytically removed during maturation. Is there any sequence of events known for BPP1? If not, that statement should be softened.

---

## [Author Response]

We are grateful to Prof. Stephen Harrison, Prof. NikoGriegorieff, and the anonymous third reviewer for their in-depth reviews and for providing very valuable suggestions for us to improve our paper. Following their suggestions, we have carefully revised our paper and addressed all of their concerns, as detailed below. Our major changes include the following:

1) We have performed additional verification of chain tracing by forcing the HK97 topology into our cryoEM map. We made new figures based on this chain tracing results (Figure 4 and Figure 4—figure supplement 3) and added new text to the Results and Materials and methods sections.

2) We have removed the discussion on evolution and incorporated all the texts suggested by the reviewers.

3) One reviewer asked for more detailed information about data processing and we have expanded our Materials and methods section to include more details.

4) We deleted the previous Figures 2, 4 and 6, Figure 4—figure supplement 1, and Figure7–figure supplement 2A-D, and included some new panels, such as Figure 2 and Figure 4, as suggested by reviewers.

5) We have made great effort to condense our text and have shortened the Introduction, Results, and Discussion sections.

*1) The authors misunderstand and misstate the history of discovery and understanding of viral capsid subunit structure*.

Thank you for correcting our misunderstanding of the early history of virus crystallography and for proposing the two replacement paragraphs. We have replaced the first three paragraphs of the original Introduction by the two short paragraphs and added appropriate references.

*2) They need to provide more details about data processing*.

We have expanded our Materials and methods section and the requested details are now included.

We have also now included FSC and R-factor curves in the revised Figure 2.

*3) They confuse structural observations with energetic ones, so their speculations about “withstanding high pressures” (moreover, is 40 atm really that high, on a molecular scale?) are misplaced*.

We have removed the description about the high pressure in the revised manuscript.

There appears to be a difference in personal preference in the use of terminology here. Chainmail is an extended fabric composed of catenated rings (i.e*.*, topological interlinks). It must have rings, and we show in the paper that neighboring MCP subunits form rings (old Figure 7 and new Figure 6). We also show that the rings are catenated, thus meeting the topological requirement for chainmail.

In the face of this evidence, the reviewers may choose not to apply the term chainmail, or not to apply the description noncovalent chainmail. We respect that position, but we choose to take the opposite. We recognize that chainmail was first used by Duda and Johnson et al. to described catenated rings observed in HK97. The catenated rings observed in BPP-1 have the same topology as that in HK97, except that the MCP subunits within each ring interact non-covalently instead of covalently as in HK97. We find that non-covalent chainmail serves to capture the key elements of the complex nature of the molecular interactions we observe in the BPP-1 and other phages. One can also apply the description of noncovalent chainmail to what others have observed in a number of bacteriophages (see, for example, Figure 3 of Lander et al.). We expect that the field will benefit by more widespread adoption of this readily grasped term, which greatly simplifies the description of the rather complex structural organization and facilitate comparison.

*4) They need to present some further validation of the difference between BPP-1 and HK97 and to describe the chain trace and the overall structure more clearly. The current description makes it impossible for anyone except the initiated to understand the interesting relationship between BPP-1 MCP and the HK97 coat protein, unless the reader realizes that all they need do is look at*
Figure 6*. Moreover, the description of structural “blocks” is misleading, as the two strands in the central sheet and the long beta hairpin are parts of distinct structural modules*.

Indeed, we were 100% sure that our chain trace is correct on the basis that the majority of side chain densities are clearly resolved in our map and they match those in our model. Because many of the 20 amino acid have distinguishable side chain features in our cryoEM map, this matching for a protein of more than 300 residues provides an extremely strong constraint and virtually eliminates any possibility of incorrect chain tracing.

Nonetheless, we now performed the additional verification of our model as suggested by this reviewers. In this verification modeling experiment, we swapped the α andβ structural elements in our de novo BPP-1 MCP model to create an interchanged model that matches the HK97 topology. This interchanged modelwas then refined with Phenix for five cycles. Despite this refinement effort, the final interchanged model does not agree with our experimental cryoEM density in many ways. We made a new figure (Figure 4—figure supplement 3) that show some examples of these disagreements: 1) panels Band Cshow the disagreements of the cryoEM density map and two secondary structural elements, βG and α5, which are directly connected to βL in BPP-1 and HK97, respectively; 2) at the two interchanging positions, there are no EM densities corresponding to the linking loops in this interchanged model (panel A); 3) there is continuous cryoEM density extending beyond the C-terminus of the interchanged model (blue arrow in panel A); 4) most importantly, many side chains in the interchanged model do not match those resolved in the cryoEM density map (all panels, some details shown in panels Band C).

We have now also changed the term “building block” to “structural element” throughout.

We agree with the reviewers that “interchange” also accurately describes the nature of the topological differences observed. However, the term “non-circular permutation” has been extensively used in the literature to describe such differences among proteins. We decided to follow what is in the literature.

*5) They should tone down evolutionary speculations. (Francis Crick once said something to the effect that we should always imagine evolutionary scenarios and talk about them, to sharpen our thinking, but we shouldn't publish them, unless there is good evidence.*)

We have removed text from the revised version of the manuscript to address this point.

We have also clarified in the revision that the jellyroll lineage in this paper includes both single- and double-jellyroll motifs.